# Antibody responses to Japanese encephalitis virus and dengue virus serotype 2 in children from an orthoflavivirus endemic region after IMOJEV vaccination

Fatima Ericka S. Vista[1,2,3]*, Leslie Michelle M. Dalmacio[1], Pauline R. Solis[4], Cecilia Nelia C. Maramba-Lazarte[5,6], Diane M. Lang[2,3], Alan L. Rothman[2,3], Sheriah Laine M. de Paz-Silava[7]

1 Department of Biochemistry and Molecular Biology, College of Medicine, University of the Philippines Manila, Manila, Philippines, 2 Institute for Immunology and Informatics, University of Rhode Island, Providence, Rhode Island, United States of America, 3 Department of Cell and Molecular Biology, University of Rhode Island, Providence, Rhode Island, United States of America, 4 Department of Pediatrics, Cagayan Valley Medical Center, Tuguegarao City, Cagayan, Philippines, 5 Department of Pharmacology, College of Medicine, University of the Philippines Manila, Manila, Philippines, 6 Institute of Herbal Medicine, National Institutes of Health, University of the Philippines Manila, Manila, Philippines, 7 Department of Medical Microbiology, College of Public Health, University of the Philippines Manila, Manila, Philippines

* fsvista2@up.edu.ph

## Abstract

### Background

Japanese encephalitis virus (JEV) is a mosquito-borne pathogen that causes severe neurologic disease. Its endemicity in Asia has prompted its inclusion in nationwide immunization programs. However, the Philippines, which is also endemic for related viruses like dengue (DENV), has not yet adopted this practice. Vaccine hesitancy is a major challenge, exacerbated by concerns over cross-reactive antibodies that may enhance viral infection. This study aimed to determine whether IMOJEV vaccination would induce cross-neutralizing or enhancing antibodies against DENV.

### Methodology/Principal findings

Pre- and one-month post-vaccination samples from IMOJEV-vaccinated Filipino children (9–24 months old) were analyzed. A reporter virus particle (RVP)-based neutralization assay against JEV showed neutralization in 28/29 subjects post-vaccination. Presence of DENV2-reactive antibodies was measured via DENV2 VLP ELISA, which revealed increased DENV2 binding reactivity post-vaccination. Pre-vaccination DENV2 binding reactivity also had no significant correlation with the JEV vaccine response. RVP-based neutralization and enhancement assays against DENV2 showed that there was no significant change in neutralizing or enhancing antibody activity against DENV2 after JEV vaccination.

**Data availability statement:** Data Availability Statement: All relevant data are within the manuscript and its Supporting information files, with the exception of exact participant age, which is potentially identifying. Access to this data is restricted by the University of the Philippines Manila Research Ethics Board (UPMREB) to protect participant confidentiality. Data may be made available upon reasonable request to the UPMREB (upmreb@post.upm.edu.ph).

**Funding:** This project was done as part of FESV's Fulbright fellowship. We gratefully acknowledge the support of the Fulbright U.S. Student Program, which is sponsored by the U.S. Department of State and the Philippine-American Educational Foundation (PAEF). The funders had no role in study design, data collection and analysis, decision to publish, or preparation of the manuscript. The contents are solely the responsibility of the authors and do not necessarily represent the official views of the Fulbright Program, the U.S. Department of State, or PAEF.

**Competing interests:** I have read the journal's policy and the authors of this manuscript have the following competing interests: ALR has served as a consultant to Takeda, Inc. These competing interests will not alter adherence to PLOS policies on sharing data and materials. All other authors have declared that no competing interests exist.

## Conclusions/Significance

This study shows that IMOJEV vaccination elicited a JEV neutralizing response in 97% of vaccinees and that the magnitude of JEV neutralizing titers post-vaccination was not associated with pre-existing binding antibodies to DENV2. Further, while live JEV vaccination increases DENV2-binding antibodies, this cross-reactivity does not lead to DENV2 enhancement. These findings contribute to a better understanding of the orthoflavivirus antibody response following immunization and the influence of pre-existing heterologous orthoflavivirus antibodies. This could guide vaccination strategies, especially in orthoflavivirus-endemic regions.

## Author summary

Japanese encephalitis virus (JEV) is the leading pathogen causing viral encephalitis in Asia. Fortunately, there are licensed JEV vaccines incorporated into the routine immunization programs of many Asian countries. In the Philippines, this practice has not yet been adopted on a national scale. This is partly due to concerns over cross-reactive antibody interactions with dengue virus (DENV), a closely related virus that is also endemic in the region. This can happen in a phenomenon known as antibody-dependent enhancement (ADE), which was demonstrated in a previous DENV vaccine trial in the country. To address this, we measured the antibodies against DENV2 and JEV before and after vaccination in children who were administered a live JEV vaccine. We found that most children were able to neutralize JEV post-vaccination, and while this led to increased DENV2 cross-reactivity, the antibodies did not increase DENV2 enhancement *in vitro*. Our findings suggest that live JEV vaccination is unlikely to increase the risk of severe dengue, at least in the short term. Understanding cross-reactive antibody interactions is crucial for designing vaccines and in shaping vaccine strategies, which are especially relevant in endemic areas.

## Introduction

Orthoflaviviruses are a group of single-stranded RNA viruses usually transmitted by arthropod vectors. This virus family includes global public health threats like dengue virus (DENV), Japanese encephalitis virus (JEV), Zika virus (ZIKV), West Nile Virus (WNV), and Yellow fever virus (YFV) [1,2]. DENV, JEV and ZIKV, in particular, are known to be prevalent in tropical countries like the Philippines [3–5].

Japanese encephalitis virus (JEV), is a virus transmitted by *Culex* mosquitoes that can cause severe neurologic disease in infected individuals [6]. In Asia, it is the main cause of viral encephalitis [7]. While 99% of individuals infected via mosquito bites remain asymptomatic, the condition is fatal for a third of symptomatic patients. Meanwhile, survivors often deal with long-term neurologic sequelae [6]. Due to the severe

nature of the disease, there have been many efforts towards JEV vaccine development since the 1980s. Several vaccines are available, with early ones being inactivated mouse brain-derived or Vero-cell derived vaccines using the Nakayama or Beijing strains. The newer vaccines use the attenuated SA-14-14-2 strain, with the inactivated vaccine mainly used in the United States and Europe, while the live vaccines are more commonly used in Asia [8]. These have been incorporated into the national immunization programs of many Asian countries, such as Japan, China, Korea, Taiwan, and Thailand [9,10]. In the Philippines, JEV is known to be prevalent, so vaccination is recommended by the Philippine Pediatric Society for children at least 9 months of age [11]. However, the vaccine is still not a part of the nationwide routine immunization program [4,10].

DENV, on the other hand, is transmitted by *Aedes* mosquitoes and is the most prevalent flaviviral illness worldwide, with over 100–400 million infections annually [12]. It causes non-specific flu-like symptoms such as fever, headache, rashes, and joint pains, from which patients typically recover through supportive care. However, in severe cases that usually happen during a secondary DENV infection, more severe hemorrhagic manifestations are observed and may even lead to death [12,13]. Vaccines such as Dengvaxia and Qdenga have been developed for this disease [14], but neither are currently licensed for use in the Philippines.

In this study, we explore two important concepts in orthoflavivirus immunization: the role of pre-existing antibodies and antibody-dependent enhancement. In children, pre-existing orthoflavivirus antibodies may be from maternal antibodies or early natural infection. For other diseases, it is believed that maternal antibodies may interfere with the vaccination response due to immunologic blunting [15–19]. It is proposed that this happens when maternal antibodies dampen the infant's immune response to vaccination through mechanisms such as live virus neutralization or epitope masking [20]. Since the only licensed orthoflavivirus vaccines that are given routinely in childhood are for JEV and YFV [14,21] and because this practice is only done in endemic regions, studies on the influence of maternal antibodies on orthoflavivirus vaccination are limited. Meanwhile, research on heterologous orthoflavivirus infections has shown that, due to the antigenic similarities between these viruses, previous exposure affects the immune response to a subsequent orthoflavivirus encounter. The nature of this effect depends on factors such as the order, timing, and kind (whether natural or vaccine-induced) of exposures; differences may also be found between vaccine types. The existing literature on this topic has been extensively reviewed elsewhere [22,23].

Antibody-dependent enhancement (ADE), on the other hand, is a phenomenon wherein cross-reactive antibodies present at low levels are not capable of neutralizing the virus. In turn, the binding facilitates virus entry and enhancement of infection, which can contribute to severe disease [24]. Health officials in the Philippines have become wary about this due to the country's previous experience with the Dengvaxia vaccine, wherein clinical trial data for the vaccine showed that children who were DENV-naïve upon immunization had an increased risk of developing severe dengue when exposed to the virus [25]. Parents have also become cautious of vaccination campaigns after the Dengvaxia controversy [26–29], and the importance of this phenomenon has initiated its investigation of other existing orthoflavivirus vaccines [30]. For JEV, the current evidence has largely been mixed with key differences in participant recruitment and vaccine regimens between studies, making comparisons difficult [31–34]. Further, the previous literature has primarily been focused on inactivated vaccines and on old JEV vaccine strains which are quickly being replaced by live SA-14-14-2 vaccination in Asia [10,35,36].

Given the public health significance of Japanese encephalitis and the need for data to guide vaccination policies in regions that use this vaccine, our group sought to determine if pre-existing anti-DENV2 antibodies would affect the neutralizing titers against JEV post-vaccination and whether immunization with a live attenuated JEV vaccine strain would lead to an increased tendency toward ADE of DENV2 infection in Filipino children. We found that baseline DENV2 binding reactivity was not associated with the JEV vaccine response and observed no significant vaccine-induced DENV2 cross-protection or enhancement in our cohort.

## Methods

### Ethics statement

The samples used in this work came from a cohort study led by investigators from the University of the Philippines Manila. There were a total of 40 children in the original cohort whose guardians or parents all provided written informed consent. Of the 40 participants, 29 gave consent for future use of biobanked sera. The cohort study was approved by the University of the Philippines Manila Research Ethics Board (UPMREB 2021-0739-01). The use of biobanked sera and access to demographic data for this present study was approved by the Institutional Review Board at the University of Rhode Island (IRB Reference #2225128-1) and the University of the Philippines Manila Research Ethics Board (UPMREB 2024-0613-01).

### Serum samples and IMOJEV vaccine

Biobanked sera from 29 children who were 9 months to 2 years old upon vaccination with IMOJEV were used in this study. IMOJEV is a monovalent, chimeric JEV vaccine that uses the yellow fever 17D-204 virus as its backbone, with the premembrane (prM) and envelope (E) sequences replaced by that of the live-attenuated JEV SA-14-14-2 vaccine virus, which was generated from the wild type SA14 strain by serial passaging [37]. Pre- and 1 month-post vaccination samples were available for each child for a total of 58 samples. Recruitment and vaccination were done from July - December 2022 in Tuguegarao City, Cagayan, and Cabagan, Isabela. These are municipalities in the Philippines found in the Cagayan Valley Region, known to be endemic for both JEV and DENV. None of the children who participated in the study received other flavivirus vaccines (i.e., dengue or yellow fever).

### DENV2 and JEV VLP IgG ELISA

ELISA assays were performed as previously described [38]. Briefly, clear 96-well flat-bottom plates were coated overnight at 4°C with 10 ng/well of DENV serotype 2 virus-like particle (VLP) or 25 ng/well of JEV VLP (The Native Antigen Company) diluted in coating buffer (Sigma-Aldrich). This was followed by one round of washing using 1X phosphate-buffered saline (PBS) with 0.05% Tween 20 and blocking for 1 hour at 37°C with blocking buffer (Sigma-Aldrich). After blocking, the plates were washed three times. Then, sera diluted to 1:200 in blocking buffer was added to each well and incubated at 37°C for 1 hour. Plates were washed five times before adding goat anti-human IgG with conjugated HRP (Bethyl Lab) diluted 1:25,000 in blocking buffer. This was incubated for 30 minutes at 37°C. After five washes, the detection was done using 1-Step TMB ELISA Substrate Solution (Thermo Scientific). Ten percent 2N sulfuric acid diluted in deionized water was added after 12 minutes to stop the reaction and read at 450 nm and 560 nm. Each serum was assayed in duplicate, and $OD_{450} - OD_{560}$ values were averaged. A serum sample known to be negative for DENV IgG (LGC Clinical Diagnostics) was used as a negative control.

### Cells

K562 and Raji-DCSIGNR cells were grown in RPMI-1640 Medium, while HEK-293T cells were grown in DMEM, high glucose (HyClone Laboratories Inc.). Both cell culture media were supplemented with 10% heat-inactivated fetal bovine serum (HyClone Laboratories Inc.), 100 U/mL penicillin, and 100 µg/mL streptomycin (Sigma-Aldrich). Cells were maintained at 37°C with 5% $CO_2$.

### Generation and titration of reporter virus particles (RVP)

Reporter virus particles were generated by co-transfecting in HEK-293T cells a green fluorescent protein (GFP) tagged West Nile Virus (WNV) replicon plasmid and a structural protein cassette expression plasmid (CprME) for DENV2 strain 16681 (GenBank accession no. NC_001474) or JEV strain 7812474 (GenBank accession nos. EF688633 and U70387)

(kind gifts from Dr. Stephen Whitehead and Dr. Gregory Gromowski) as previously described [39,40] with some modifications. Transfection of HEK-293T cells was performed in T75 flasks using 32 μg of DNA (replicon plasmid to CprME plasmid ratio of 1:3) and Lipofectamine 3000 (ThermoFisher Scientific). Cells were incubated for 3 days (DENV2) or 5 days (JEV) at 30°C with 5% $CO_2$ before harvesting the RVPs. Upon harvest, the supernatant was clarified by centrifugation, and aliquots were stored at −80°C prior to use. Titration was performed by adding two-fold serially diluted virus particles to 5 x $10^4$ Raji-DCSIGNR cells and incubating for 42 hours at 37°C. Infection was stopped by adding BD Cytofix Fixation Buffer (BD Biosciences) diluted 1:5 in 1X PBS after two rounds of washing with 1X PBS. The number of GFP-positive cells was measured via flow cytometry using a MACSQuant Analyzer 10 (Miltenyi Biotec), with GFP detected at 488 nm excitation and 525/50nm emission.

### RVP neutralization and enhancement assays

Sera were heat-inactivated at 56°C for 30 minutes prior to use. Starting at a dilution of 1:10, three-fold (neutralization) or four-fold (enhancement) serial dilutions of the sera were prepared and incubated for 1 hour at 37°C with an equal volume of solution containing JEV or DENV2 RVP diluted to a concentration yielding <10% infection on titration as described above; the same RVP dilution was used in enhancement assays. The RVP-antibody mixture was then added to 5x$10^4$ Raji-DCSIGNR (neutralization) or K562 (enhancement) cells and incubated at 37°C for 42 hours. Fixation and flow cytometry were done as detailed above.

### Statistical analyses

For the ELISA assays, positive/negative (P/N) values were calculated by dividing the OD value of each sample by the OD value of the negative control. The cut-off setting was done by identifying all test subjects with a P/N value ≤ 1.0. The standard deviation (SD) for their samples and the negative control was multiplied by three (3SD), and all values above a P/N of 1.0 + 3SD were considered reactive.

Flow cytometry data were analyzed using FlowJo V10.4.2. The cell population of interest was identified and gated using the forward scatter area (FSC-A) and side scatter area (SSC-A) parameters based on cell size and granularity. Gating for positive GFP expression was done using uninfected controls as the baseline. For neutralization assays, the 50% RVP neutralization titer (RVPNT50) was determined by first normalizing the data against medium-only and virus-only controls. With this data, a non-linear regression analysis using the log(inhibitor) vs. normalized response -- variable slope equation was performed. Samples were considered to have neutralizing activity at RVPNT50 ≥ 10. Results were presented as $\log_{10}$ values of the RVPNT50 titers with a value of 5 assigned to titers < 10. The fold enhancement on the antibody-dependent enhancement (ADE) assay was derived by comparing the values with serum in a virus-only control. Samples were considered to have enhancing activity when peak enhancement was ≥ two-fold compared to the control.

Comparisons between groups were done using the Student's t-test, and correlation analyses were done using the Spearman rank correlation test. Results were considered statistically significant for a p-value < 0.05. All statistical analyses were carried out using GraphPad Prism version 10.3.1.

## Results

### Baseline DENV2 and JEV binding reactivity in Filipino infants prior to IMOJEV vaccination

To measure the baseline presence of DENV2 antibodies in children prior to vaccination, their binding reactivity was evaluated using a DENV2 VLP IgG ELISA. We compared binding reactivity to DENV2 in children less than or greater than 12 months of age since studies have shown that maternal antibodies would have already waned in most children by this age [41,42]. Of the 29 children, 13 were less than 12 months old, while 16 were 12 months or older upon vaccination. At the set cut-off, 3/13 children from the younger group were considered DENV2-reactive. In contrast, there were 5/16

DENV2-reactive subjects from the older group (Fig 1A), giving an overall DENV2 seroprevalence of 28% that suggests most children in this cohort were DENV2-naïve upon vaccination. There was no significant difference (p = 0.16) between the DENV2 binding reactivities of the two groups. DENV2 binding reactivity also showed a weak non-significant (p = 0.06) positive association (r = 0.36) with age (Fig 1B). The baseline reactivity to JEV VLP was also measured, and 7/8 of the DENV2-reactive subjects were also found to be JEV-reactive (Fig 1C). There was no significant difference in JEV reactivities between groups (p = 0.42), and no significant (p = 0.14) correlation (r = 0.28) was found between JEV reactivity and age (Fig 1D). Further analysis showed a positive correlation between the two assays with a Spearman r of 0.78 (p-value <0.0001) (Fig 1E). The findings suggest that the JEV VLP IgG ELISA is not highly specific for JEV reactivity and instead reflects the binding of cross-reactive DENV2 antibodies.

### JEV binding reactivity and neutralization after IMOJEV vaccination

Next, vaccine response was characterized through an RVP neutralization assay against JEV. None of the children showed neutralization activity against JEV prior to vaccination (S1 Fig). At 1 month post-vaccination, there was a significant increase in JEV neutralizing antibody titer in a majority of the children (p < 0.0001), with 28/29 (97%) having RVPNT50 titers ≥ 10 (geometric mean titer: 254.7) (Fig 2A). Only subject B27 failed to respond to vaccination. The JEV IgG binding reactivity also showed a significant increase compared to baseline (p = 0.001) (Fig 2B). However, the ELISA was only able to detect JEV reactivity in half (14/28) of the vaccinees, showing the poor sensitivity of this assay for vaccine-induced antibodies at 1 month post-vaccination. Additionally, 7 out of the 14 subjects were already JEV-reactive pre-vaccination despite having no JEV-neutralizing activity, and an increase in reactivity to JEV was also seen for subject B27 despite the lack of a neutralizing antibody response. Four subjects (B12, B24, B37, B40) were noted to have decreased ELISA reactivity to JEV post-vaccination.

We also explored whether the magnitude of vaccine response was associated with age or pre-vaccination DENV2 binding reactivity. We found no significant differences between age groups (Figs 2C and S2A; p = 0.35 and 0.24, respectively). The geometric mean titer of JEV-neutralizing antibodies post-vaccination was lower for the DENV2-reactive group than the non-reactive group (324 vs. 135.3, respectively), but this difference was not statistically significant (p = 0.11, Fig 2D). Pre-vaccination DENV2 binding reactivity also had a weak non-significant (p = 0.09) negative correlation (r = -0.32) with JEV titers post-vaccination (S2B Fig).

### DENV2 binding reactivity and functional antibody responses after IMOJEV vaccination

Next, antibody responses to DENV2 were evaluated in 17 subjects that were selected based on age, with 9 from the younger (<12 months) age group and 8 from the older (≥12 months) age group. Among these 17 children, 7 were DENV2-reactive at baseline, while 10 were not. Fig 3A shows that, like JEV VLP binding reactivity, DENV2 VLP binding significantly increased 1 month after immunization in most subjects (p = 0.003). Six children (B12, B24, B38, B35, B37, B39) had decreased reactivities post-vaccination. Consistent with the previous observation on the relationship between DENV2 and JEV ELISA results, the fold change in P/N values after vaccination was also positively correlated between the two assays with a Spearman r of 0.62 (p = 0.0002) (Fig 3B).

To determine if the DENV2-reactive antibodies in these children are also neutralizing, an RVP neutralization assay against DENV2 was performed. The DENV2 serotype was chosen as the focus of this study due to its prevalence in the Philippines and its relevance to ADE [43,44]. ELISA assays against DENV 1, 3 and 4 VLP antigens also showed a high degree of correlation (Pearson r = 0.66 to 0.78) (S3 Fig) with DENV2 antibody binding. In this set, only 4 out of 7 DENV2-reactive subjects showed neutralization against DENV2 post-vaccination (S4 Fig). Three of them (B01, B10, B27) also showed pre-vaccination neutralizing activity against DENV2 (Fig 3C); all were from the older age group, which further supports that these antibodies are likely from natural DENV exposure. Of the four children, three were vaccine responders

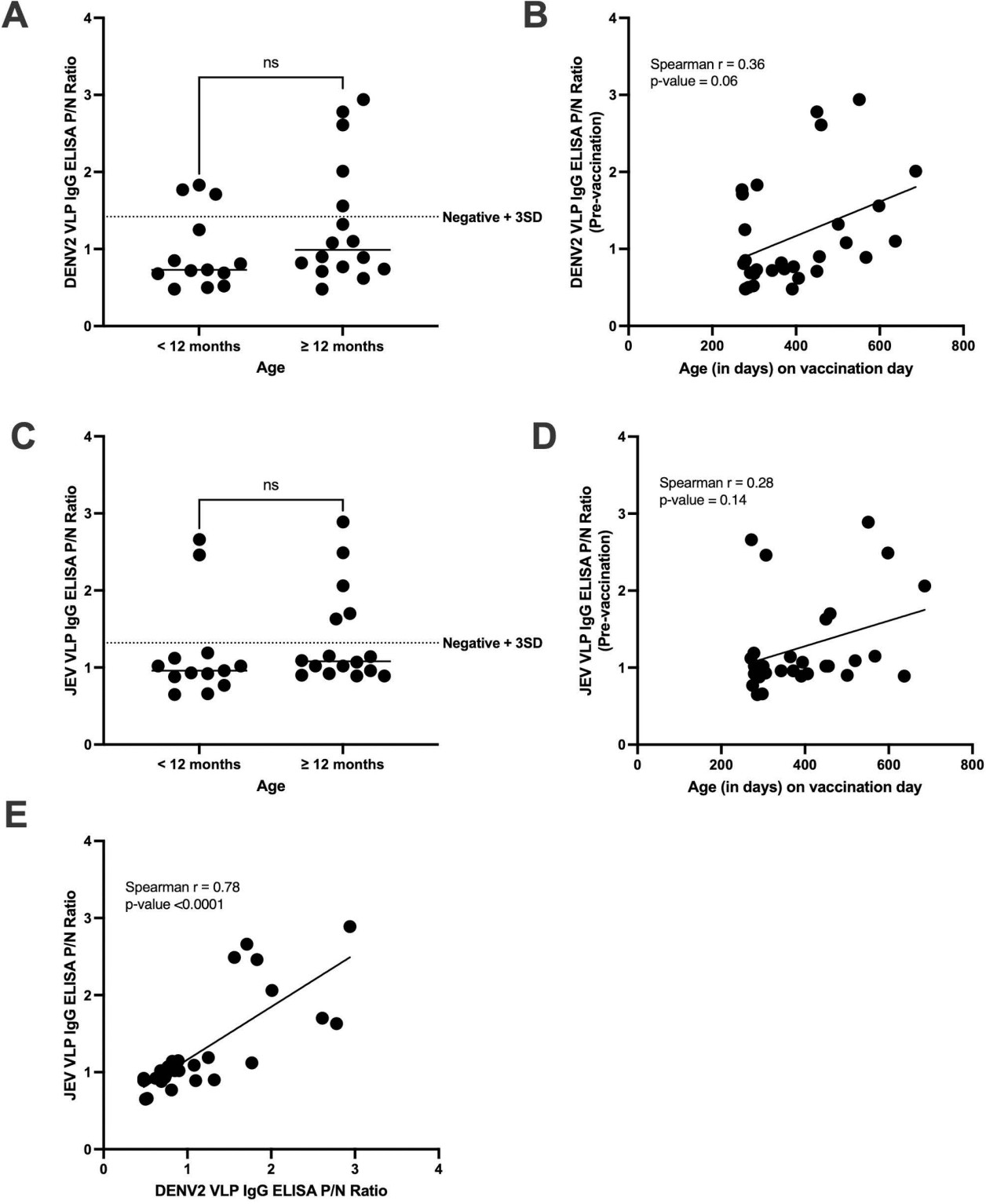

**Fig 1. Baseline IgG binding to DENV2 and JEV VLP. (A and C)** Binding reactivity to (A) DENV2 or (C) JEV of children younger than 12 months versus children 12 months and older prior to vaccination (two-tailed Student's t-test; broken lines at P/N cut-off; horizontal bars represent the mean). **(B and D)** Relationship between age and binding reactivity against (B) DENV2 or (D) JEV (two-tailed Spearman rank correlation test). **(E)** Relationship between DENV2 and JEV binding reactivities (two-tailed Spearman rank correlation test).

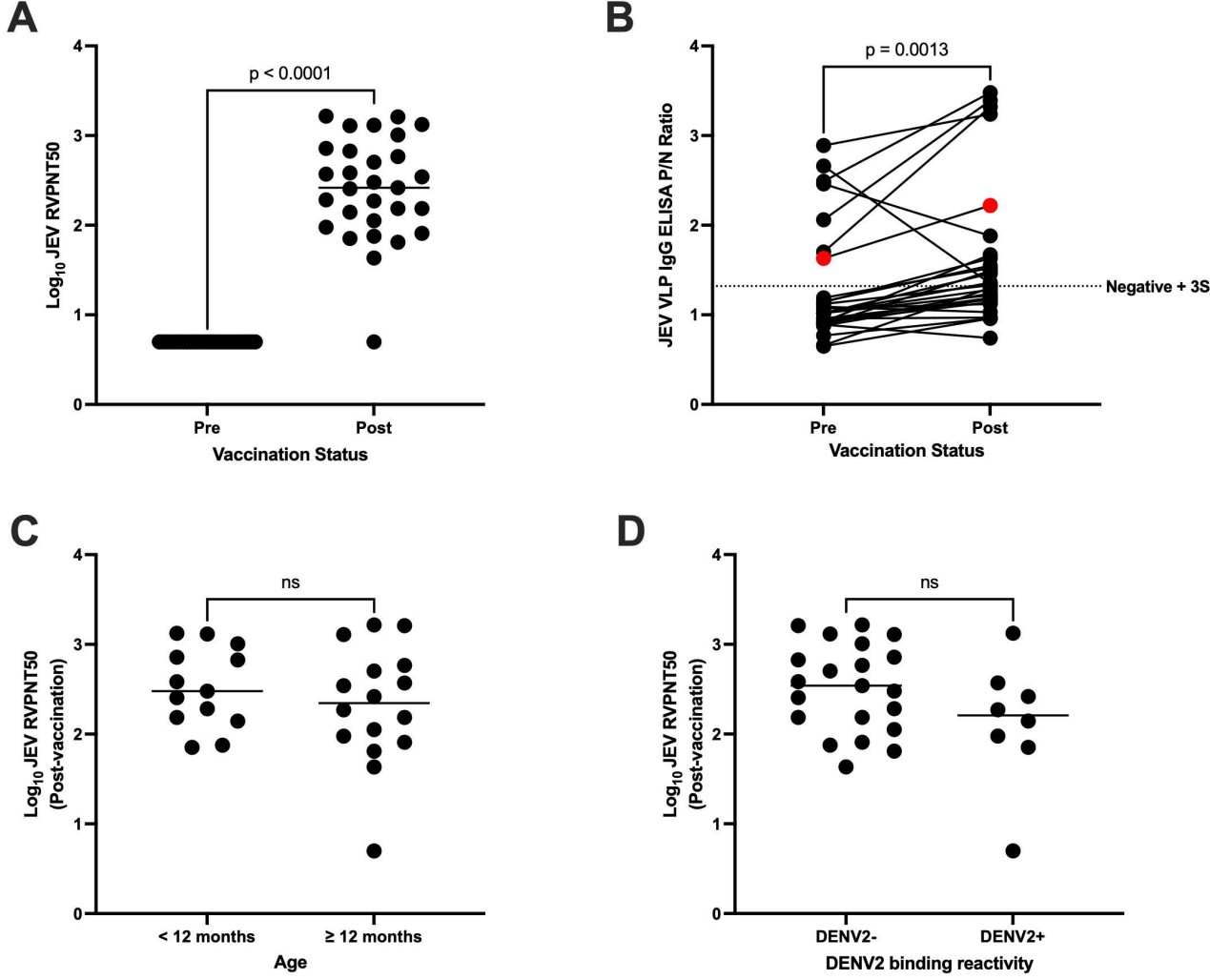

**Fig 2. Antibody response to IMOJEV vaccination. (A)** Comparison of $\log_{10}$ JEV RVPNT50 titers pre- and post-vaccination (one-tailed paired Student's t-test; titers less than 10 were assigned a value of 5). **(B)** Pairwise comparison of JEV VLP IgG binding reactivity pre- and post-vaccination (one-tailed paired Student's t-test; in red is subject B27, who did not respond to vaccination based on the RVP neutralization assay). **(C and D)** Comparison of post-vaccination $\log_{10}$ JEV RVPNT50 titers (C) between age groups and (D) by DENV2 reactivity (two-tailed unpaired Student's t-test).

(B01, B10, B24), and their DENV2 neutralization titers increased slightly post-JEV immunization (Fig 3C). Subject B27, who did not develop JEV-neutralizing antibodies, showed a decline in DENV2-neutralizing antibodies after 1 month.

Since half of the DENV2-reactive children did not show DENV2 neutralization, we hypothesized that the observed binding is from subneutralizing antibodies that may enhance DENV infection. We also wanted to know whether JEV vaccination would increase any observed enhancement compared to pre-vaccination sera. To evaluate this, an *in vitro* DENV2 RVP ADE assay in K562 cells was performed. Six out of the 7 DENV2-reactive children also enhanced DENV2 infection *in vitro* (S5 Fig). However, no significant change in enhancement activity was found post-vaccination (p = 0.27) (Fig 3D and Table 1). Subjects with enhancing but non-neutralizing antibodies to DENV2 (E + /N-) prior to vaccination tended to have the highest neutralizing antibody responses to JEV post-vaccination, but the differences were not statistically significant (S6 Fig). A master table of the ELISA, neutralization and enhancement assay results can be seen in S1 Table.

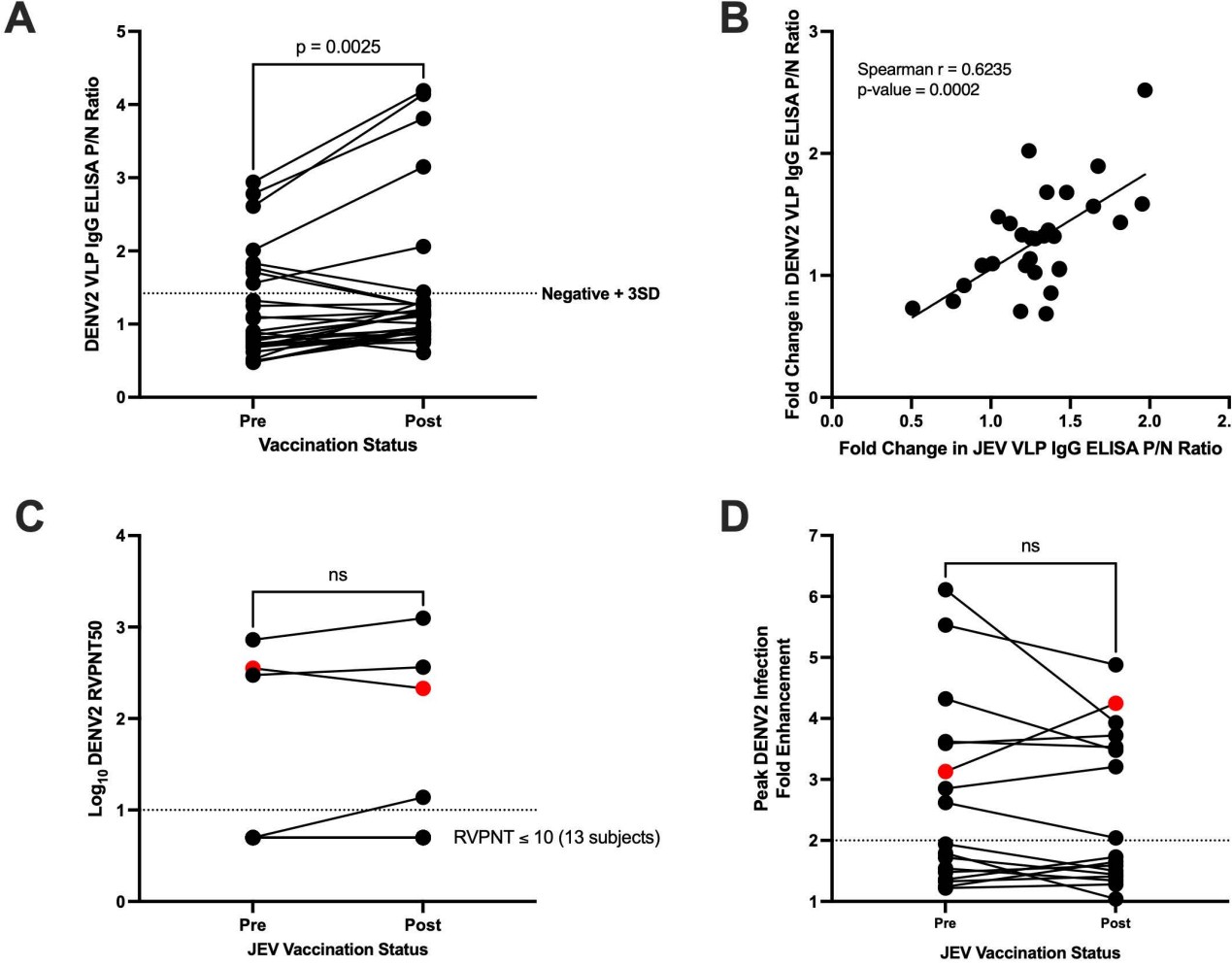

**Fig 3. DENV2 binding reactivity and functional antibody responses. (A)** Pairwise comparison of DENV2 VLP IgG binding reactivity pre- and post-vaccination (one-tailed paired Student's t-test). **(B)** Relationship between the fold change in DENV2 and JEV VLP IgG ELISA P/N values after vaccination (one-tailed Spearman rank correlation test). **(C and D)** Pairwise comparison of DENV2 neutralizing titers or fold enhancement post-JEV immunization (two-tailed paired Student's t-test; broken lines at cut-off for neutralization or enhancement and shown in red circles is subject B27 who did not respond to JEV vaccination; in Fig 3C, the line at y = 0.7 represents 13 subjects that did not have neutralizing activity pre- and post-vaccination).

## Discussion

Reports of severe dengue after DENV vaccination have led to increased vaccine hesitancy among parents in the Philippines [26–29]. This may be one of the reasons why earlier efforts to include JEV in the national immunization program have since stalled. Meanwhile, cases of JEV continue to plague the country, with an average annual incidence rate of 1 per million total population since 2008, according to the World Health Organization (WHO) [45]. This is much higher than neighboring countries with routine immunization programs, such as Japan (0.03), South Korea (0.3) and Thailand (0.45). Such statistics underscore the need for routine JEV immunization for Filipino children. Here, we report that children from a JEV- and DENV-endemic region in the Philippines had strong responses to JEV vaccination regardless of their pre-existing DENV2 binding antibodies and that vaccination elicited cross-reactive antibodies to DENV2, but these did not enhance DENV2 infection *in vitro*.

**Table 1. Serum dilution at peak fold enhancement of DENV2.**

| Participant ID | Pre-vaccination | Post-vaccination |
| --- | --- | --- |
| B01 | 1:160 | 1:160 |
| B06 | 1:10 | 1:40 |
| B10 | 1:640 | 1:640 |
| B12 | 1:10 | 1:10 |
| B15 | 1:10 | 1:10 |
| B19 | 1:40 | 1:10 |
| B22 | 1:40 | 1:10 |
| B23 | 1:10 | 1:10 |
| B24 | 1:10 | 1:10 |
| B25 | 1:10 | 1:10 |
| B27 | 1:160 | 1:640 |
| B31 | 1:10 | 1:40 |
| B32 | 1:40 | 1:40 |
| B34 | 1:40 | 1:40 |
| B36 | 1:10 | 1:10 |
| B37 | 1:10 | 1:10 |
| B38 | 1:10 | 1:10 |

In this cohort, we found a baseline DENV2 seroprevalence of 28%. The lower seropositivity rate in our study is consistent with the observation that DENV seropositivity tends to increase in older age groups [46] and stresses the burden of orthoflavivirus exposure in the region. We also measured the baseline IgG seroprevalence against JEV using ELISA but found that the results reflected DENV cross-reactivity rather than true JEV binding, highlighting the poor performance of this assay in detecting JEV vaccine response similar to what other groups have observed [47–49]. Our preliminary testing on a subset of samples also showed that vaccine response could not be detected at antigen concentrations of 10, 25 and 50 ng/well (S2 Table). A correlation matrix between the DENV2 and JEV ELISA binding reactivity values with neutralization titers also showed that DENV2 correlation is high for both pre- and post-vaccination samples (Pearson r = 0.74 to 0.76), but this is not the case for JEV (S7 Fig). The presence of vaccine-induced anti-JEV IgM in the serum may also explain the findings. In previous studies, it was shown that 30–40% of vaccinees still had detectable IgM levels one month post-vaccination which may interfere with IgG detection [49,50].

Analyzing the relationship between DENV2 reactivities and age, we found a positive correlation between the two. In older children, this likely reflects natural DENV exposure rather than maternal DENV antibodies as supported by finding DENV neutralizing antibodies only in older children. However, the weak and non-significant correlation also implies that maternal antibodies may still be present in some children. In infants, DENV antibodies are usually expected to decrease after birth due to the waning of maternal antibodies, but these may already be undetectable in older infants [41,42,51].

The neutralization and ADE assays were performed using an RVP-based platform previously developed by others [39,40]. The use of RVPs allows for high-throughput and rapid evaluation of functional antibody responses to viruses while being relatively safer compared to the traditional plaque assay due to the non-replicating nature of the pseudoinfectious particles [52,53]. In this assay, RVP infection of cells is measured via the optical detection of a GFP-tagged reporter gene expressed by the RVPs [40,52]. For JEV, neutralizing titers ≥ 10 on a plaque reduction neutralization test (PRNT) are considered seroprotective [54]. In this study, 97% of the children developed neutralizing titers ≥ 10 against JEV 28 days after vaccination, which is indicative of a good vaccine response, as previous groups have reported for the IMOJEV vaccine [55,56]. We note, however, that the antibody level threshold for the RVP-based neutralization assay has not yet been

validated. We also examined JEV vaccine response by age to explore if maternal antibodies, which may still be present in some younger children, would interfere with the immune response. This a phenomenon that has been observed in vaccines for diseases such as pertussis, measles and malaria [17–19]. In our study, JEV vaccine responses did not differ by age and pre-vaccination DENV2 binding reactivity was not shown to be associated with vaccine response. However, children with pre-existing DENV2 enhancing and neutralizing antibodies showed lower vaccine responses. This may be reflective of original antigenic sin, where immunologic memory blunts the immune response to a similar but non-identical antigen [57], as also previously observed for the YFV vaccine [58]. Notably, children with enhancing but non-neutralizing DENV2 antibodies had higher JEV vaccine responses, consistent with a hypothesis that cross-reactive maternal DENV antibodies may boost the immune response through enhancement of the live attenuated virus [59].

Another significant concern in orthoflavivirus vaccine response that has not yet been well-explored for this vaccine is antibody-dependent enhancement, which is a phenomenon that makes vaccine design for orthoflaviviruses challenging since cross-reactive antibodies could be neutralizing but at certain conditions, may also be enhancing [22]. Besides Dengvaxia, other orthoflavivirus vaccines have also demonstrated cross-reactive binding to related viruses, with studies showing that this could lead to both cross-protection and enhancement [30,60,61]. In this study, we observed no significant change in DENV2-enhancing activity post-vaccination.

Other literature exploring the effects of JEV antibodies on DENV enhancement has been limited, and available studies have mixed findings. A cohort from Thailand recruited between 1998–2002 observed that the presence of detectable JEV antibodies was associated with an increased risk of symptomatic DENV infection [31]. Another study of Japanese adults vaccinated from 2009-2011 also revealed that more post-vaccination subjects exhibited *in vitro* DENV ADE compared to pre-vaccination subjects [32]. However, a third study conducted in adults from Thailand recruited in 1982 concluded that pre-existing JEV antibodies did not enhance DENV2 infection [33]. Notably, the third study involved a cohort that likely developed antibodies from natural exposure since recruitment occurred prior to routine immunization in the country. Moreover, a JEV vaccine trial conducted in 1984–1985 involving a cohort of over 65,000 children in Thailand found that there was a lower number of dengue fever and severe dengue cases among children from the vaccinated group, a trend that was observed until two years post-vaccination [34]. While they did not explore enhancement, the authors state that this finding points to possible DENV cross-protection from JEV vaccination, especially since the difference in DENV attack rates between groups was highest in the months following vaccination. In the present study, live vaccination using a chimeric JEV SA-14-14-2 strain was given to children, while the previous vaccine studies used cohorts that were given JEVAX (inactivated mouse brain-derived vaccine using Nakayama strain), JEBIK-V (inactivated Vero cell vaccine using Beijing-1 strain) and in the case of the study by Hoke et al. (1988), an inactivated monovalent or bivalent vaccine containing either the Nakayama strain alone or in combination with the Beijing-1 strain. Potential differences in enhancing activity by various JEV strains were proposed by a recent paper that identified the 106th and 107th amino acids on the envelope (E) protein as ADE-inducing epitopes. Mutating these amino acids in Nakayama, Beijing-1, and P3 strains of JEV lowered the observed DENV enhancement in immunized mice [62]. Interestingly, the L107F mutation they induced in these strains is already found on the SA-14-14-2 E protein. Differences in the mode of vaccine delivery may also play a role. Still, an *in vivo* study in mice found that both inactivated and live attenuated JEV vaccines can cross-protect against DENV 1–4 with no increased mortality indicative of enhancement [63]. Other key differences that may influence study findings include JEV exposure (whether natural or from immunization), cohort age, and the recruitment region and time period.

Our study has several limitations that should be considered when interpreting these results. First, the assays on DENV were focused on one serotype and strain (DENV 2 16681). It is possible that cross-reactive antibodies may behave differently against other dengue viruses. We were also limited by the sample size of our cohort. Similar studies on other vaccination cohorts would be useful to validate the results of this study. Being a live vaccine, IMOJEV is recommended as a one-dose regimen, but booster doses may also be given, and our study did not characterize the effect of such follow-up immunization. Further, since only one-month post-vaccination samples were tested, we cannot draw firm conclusions

about the antibody responses at later time points. In this endemic population, such studies would also be confounded by orthoflavivirus exposures in the interim. Lastly, while previous studies have shown an association between *in vitro* enhancement and severe secondary dengue illness [62,64], it is important to recognize the limitations in interpreting ADE assay results and their clinical implications. More studies using other ADE models or prospective cohorts observing vaccinated children are recommended.

Japanese encephalitis virus continues to be endemic in parts of Asia, but the resulting disease is preventable through vaccination. In this study, we show that IMOJEV vaccination can elicit neutralizing titers against JEV in Filipino children and that this immunization does not result in increased *in vitro* enhancement of DENV2 infection. We believe the results support routine immunization, especially in children who live in known JEV hotspots. As orthoflaviviruses continue to circulate worldwide, concurrent with the rollout of new vaccines and routine immunization for existing vaccines, more investigations on the orthoflavivirus immune response after natural exposure and vaccination are needed, as well as on the effect of heterologous exposures on these antibody responses.

## Supporting information

**S1 Fig. Pre- and post- IMOJEV vaccination anti-JEV neutralizing titers.**
(TIF)

**S2 Fig. Relationship between the $\log_{10}$ JEV RVPNT50 titers and (A) age or (B) DENV2 binding reactivities (two-tailed Spearman rank correlation test).**
(TIF)

**S3 Fig. Correlation between DENV2 ELISA P/N Values and DENV 1, 3 and 4 ELISA P/N Values.**
(TIF)

**S4 Fig. Pre- and post-IMOJEV vaccination anti-DENV2 neutralizing titers.**
(TIF)

**S5 Fig. Pre- and post-IMOJEV vaccination DENV2 antibody dependent enhancement of selected samples.**
(TIF)

**S6 Fig. Comparison of vaccine response based on pre-IMOJEV vaccination DENV2 functional antibody response.**
(TIF)

**S7 Fig. Correlation between (A) DENV2 or (B) JEV ELISA P/N values with neutralization titers.**
(TIF)

**S1 Table. Master table of ELISA, neutralization and enhancement assay results.**
(XLSX)

**S2 Table. Test of different JEV antigen concentrations for selected serum samples at 1:200 serum dilution.**
(XLSX)

## Acknowledgments

We wish to thank the Department of Science and Technology - Philippine Council for Health Research and Development (DOST-PCHRD) for SLMD's MECO-TECO JRP research funding, which supported the cohort study that provided samples for this study and the National Institutes of Health (NIH) for ALR's grant funding (U01 AI179523), which provided

support for the *in vitro* experiments. We thank Dr. Stephen Whitehead (US National Institutes of Health) and Dr. Gregory Gromowski (Walter Reed Army Institute of Research) for generously sharing their plasmids for use in this study as well as Dr. Day-Yu Chao for her intellectual contributions to the parent study from which this work arose. We also recognize Mr. Scott de Sagon for his technical support, and Ms. Calline Danica Gomez, Ms. Kiara Maye Sta. Ana, Ms. Edelyn Ibe and Mr. John Clarence Flores for their administrative assistance. Finally, we gratefully acknowledge the children who participated in the cohort study and their parents for providing consent.

## Author contributions

**Conceptualization:** Fatima Ericka S. Vista, Leslie Michelle M. Dalmacio, Pauline R. Solis, Cecilia Nelia C. Maramba-Lazarte, Alan L. Rothman, Sheriah Laine M. de Paz-Silava.

**Data curation:** Fatima Ericka S. Vista.

**Formal analysis:** Fatima Ericka S. Vista, Leslie Michelle M. Dalmacio, Diane M. Lang, Alan L. Rothman, Sheriah Laine M. de Paz-Silava.

**Funding acquisition:** Alan L. Rothman, Fatima Ericka S. Vista, Leslie Michelle M. Dalmacio, Pauline R. Solis, Cecilia Nelia C. Maramba-Lazarte, Sheriah Laine M. de Paz-Silava.

**Investigation:** Fatima Ericka S. Vista, Diane M. Lang, Sheriah Laine M. de Paz-Silava.

**Methodology:** Fatima Ericka S. Vista, Leslie Michelle M. Dalmacio, Diane M. Lang, Alan L. Rothman, Sheriah Laine M. de Paz-Silava.

**Project administration:** Alan L. Rothman, Sheriah Laine M. de Paz-Silava.

**Resources:** Alan L. Rothman, Sheriah Laine M. de Paz-Silava.

**Visualization:** Fatima Ericka S. Vista.

**Writing – original draft:** Fatima Ericka S. Vista.

**Writing – review & editing:** Fatima Ericka S. Vista, Leslie Michelle M. Dalmacio, Pauline R. Solis, Cecilia Nelia C. Maramba-Lazarte, Diane M. Lang, Alan L. Rothman, Sheriah Laine M. de Paz-Silava.

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
