## [Decision Letter · Decision Letter 0]

16 Mar 2025

Antibody responses to Japanese encephalitis and dengue viruses in children from a flavivirus endemic region after IMOJEV vaccination

Dear Dr. Vista,

Thank you for submitting your manuscript to PLOS Neglected Tropical Diseases. After careful consideration, we feel that it has merit but does not fully meet PLOS Neglected Tropical Diseases's publication criteria as it currently stands. Therefore, we invite you to submit a revised version of the manuscript that addresses the points raised during the review process.

Please submit your revised manuscript within 60 days May 15 2025 11:59PM. If you will need more time than this to complete your revisions, please reply to this message or contact the journal office at plosntds@plos.org. Please include the following items when submitting your revised manuscript:

We look forward to receiving your revised manuscript.

Kind regards,

Joshua Anzinger

Academic Editor

David Safronetz

Section Editor

Shaden Kamhawi

co-Editor-in-Chief

Paul Brindley

co-Editor-in-Chief

**Additional Editor Comments :**

While all reviewer comments should be addressed in full, major methodological issues were raised by two reviewers, particularly in relation to assays used to confirm DENV neutralization. There was also concern regarding the lack of assessment of other DENV serotypes. The methodological issues should be addressed with additional experimental work that provides greater strength to the author’s conclusions.

**Journal Requirements:**

1) We do not publish any copyright or trademark symbols that usually accompany proprietary names, eg ©,  ®, or TM  (e.g. next to drug or reagent names). Therefore please remove all instances of trademark/copyright symbols throughout the text, including:

- ® on page: 8

- TM on pages: 7, and 8.

2) We have noticed that you have uploaded Supporting Information files, but you have not included a list of legends. Please add a full list of legends for your Supporting Information files after the references list.

3) We notice that your supplementary figures are uploaded with the file type 'Figure'. Please amend the file type to 'Supporting Information'. Please ensure that each Supporting Information file has a legend listed in the manuscript after the references list.

4) We note that your Data Availability Statement is currently as follows: "All relevant data are within the manuscript and its Supporting Information files.". Please confirm at this time whether or not your submission contains all raw data required to replicate the results of your study. Authors must share the “minimal data set” for their submission. PLOS defines the minimal data set to consist of the data required to replicate all study findings reported in the article, as well as related metadata and methods (https://journals.plos.org/plosone/s/data-availability#loc-minimal-data-set-definition).

5) Please ensure that the funders and grant numbers match between the Financial Disclosure field and the Funding Information tab in your submission form. Note that the funders must be provided in the same order in both places as well. Currently, the order of the grants is different in both places.

**Reviewers' Comments:**

Reviewer's Responses to Questions

**Key Review Criteria Required for Acceptance?**

**Methods**

-Are the objectives of the study clearly articulated with a clear testable hypothesis stated?

-Is the study design appropriate to address the stated objectives?

-Is the population clearly described and appropriate for the hypothesis being tested?

-Is the sample size sufficient to ensure adequate power to address the hypothesis being tested?

-Were correct statistical analysis used to support conclusions?

-Are there concerns about ethical or regulatory requirements being met?

Reviewer #1: In this study, the authors intended to investigate (1) if the prior DENV infection would affect the JEV vaccination titer and (2) if the JEV vaccination-induced antibody could enhance DENV infection. The authors concluded that “while JEV vaccination increases DENV-binding antibodies, this cross-reactivity does not lead to virus enhancement”. However, the way that the data being presented caused confusion, such as presenting D2VLP-ELISA results as an indication of prior DENV exposure rather than presenting neutralization results. Furthermore, the authors conclude that “while live JEV vaccination increases DENV-binding antibody, this cross-reactivity does not lead to virus enhancement.” The only data to support such a conclusion is the enhancement of DENV-2 RVP without showing the other serotypes of DENV. The overall sample size is also too small to draw significant conclusions.

Reviewer #2: (No Response)

Reviewer #3: The methodologies described were sound and appropriately referenced.

The study is well designed, and all limitations were discussed.

**Results**

-Does the analysis presented match the analysis plan?

-Are the results clearly and completely presented?

-Are the figures (Tables, Images) of sufficient quality for clarity?

Reviewer #1: 1. The gold standard of defining the prior DENV infection is by performing neutralizing assay. However, throughout the manuscript, including the title, the authors mentioned DENV antibodies, which were actually measured only by DENV-2 RVP. This is a big concern regarding the study design and the conclusion drawn here. First, although the authors provided the D2-VLP-ELISA results, considering its poor sensitivity, how many infants having DENV exposure pre-vaccination need to be determined using all four serotypes of DENV (RVP preferred to be consistent). Secondly, it would make more sense if figure 2E and 2F presented as DENV neutralization positive or negative using all four serotypes of DENV as the indication of prior DENV infection. Third, in figure 3C-3F, it looks like the same groups of infants with pre-vaccinated DENV2 binding antibody have the neutralizing activity against DENV2 RVP and showed enhancement DENV-2 pre and post vaccination (n=8). Therefore, no statistical significance of D2 enhancement is quite clear. However, no results were shown for other serotypes of DENV. Fourth, since these infants also respond to JEV vaccination with good JEV antibody titers (except B27), the polyclonal sera include both DENV (not limited to DENV2) and JEV antibodies, performing antigen-specific IgG depletion assay by using DENV or JEV from pre- or post-vaccination sera would answer the question if JEV vaccine-induced antibody could enhance the DENV infection and vice versa.

2. Line 253-254, the authors concluded that the lack of sensitivity of JEV-VLP-based ELISA in detecting vaccine-induced antibodies. No reference has been cited or any preliminary data was performed to determine the validity of the assay, such as how much antigens are needed for coating, why 10 or 25 ng/well for JEV or D2. I wonder if the authors tried to use commercial kit for validation.

3. In this study, the authors used RVP with the backbone from WNV but the structure protein gene derived from either DENV-2 (strain 16681) or JEV (Strain 7812474). Since strain 16681 is an old strain and no longer circulates, the authors should test the enhancement of current circulation strains in the Philippines to reflect the concern of vaccine-induced severe diseases. Similarly, please provide further information if strain 7812474 is similar to the JEV circulating in the Philippines and if it belongs to the same genotype of vaccine strain SA14-14-2.

4. Did the authors acquire information about whether the recruited subjects received Dengvaxia or YFV 17D vaccination before? Such information should be revealed in the manuscript.

5. The statistical tests require further consultation with the statistician. For example, why use a one-tailed student t-test or Spearman test instead of a two-tailed one? On several occasions, when the sample size is small, the Fisher exact test should be explored.

6. Could the authors explain why only 17 subjects were selected for post vaccination DENV antibody reactivity experiments since the total 29 subjects were not too big?

7. The discussion on Line 360-367 tends to over-explain the data.

Reviewer #2: (No Response)

Reviewer #3: the results are well presented, clear and cohesive with the analysis plan

**Conclusions**

-Are the conclusions supported by the data presented?

-Are the limitations of analysis clearly described?

-Do the authors discuss how these data can be helpful to advance our understanding of the topic under study?

-Is public health relevance addressed?

Reviewer #1: (No Response)

Reviewer #2: (No Response)

Reviewer #3: The findings are described well without overstating the conclusion.

**Editorial and Data Presentation Modifications?**

Reviewer #1: 1. Throughout the manuscript, flaviruses should be corrected as orthoflaviviruses based on the recent ICTV classification

2. Line 70-71, not just DENV and JEV are prevalent in tropical countries. ZIKV as well and should be included in the sentence.

3. Line 86, DENV is “carried” should be corrected as “transmitted”

Reviewer #2: (No Response)

Reviewer #3: Nil

**Summary and General Comments**

Reviewer #1: (No Response)

Reviewer #2: The manuscript written by Vista et al. describes about analyzing antibody response patterns against DENV and JEV after immunization with attenuated-chimeric JEV vaccine. The serum samples were collected from Filipino children and were subjected to ELISA, neutralization test and ADE assay. Their results suggested that IMOJEV did not induce ADE against DENV-2 in DENV-seronegative children and did not affect the pattern of dose-response enhancing activity curves in the seropositive individuals. Authors concluded that IMOJEV were able to induce neutralizing antibody against JEV in Filipino children without induction of ADE against DENV-2. Although the sample size is small as mentioned by authors, the present information may be important for children and their parents who are concerning and doubting whether JEV vaccine might induce ADE or not. Specific comments and suggestions are described below.

In this paper, antibody levels were mostly indicated by ELISA ratio and PRNT50 titer, and both data were mixed across figures, which sometimes made it difficult to grasp the whole story. For instance, in Fig. 1C, JEV antibody levels were determined by ELISA for pre-vaccination samples, while in Fig. 2C, JEV antibody levels were determined by PRNT for post-vaccination samples. This way may not bring benefits to readers for comparing data between pre- and post-vaccination samples. Such expression needs to be improved.

I did not feel very much the necessity of ELISA data in this paper, since NT activity was demonstrated to be strongly correlated with ADE activity. To persuade the need for ELISA data, please show correlation in antibody level between ELISA ratio and PRNT50 titer when DENV antigen was used (as well JEV antigen was used).

Minor comment

Statement about Fig. 3F is missing from the manuscript.

Reviewer #3: This manuscript is very well written with good study aims and significance.

PLOS authors have the option to publish the peer review history of their article (what does this mean? ). If published, this will include your full peer review and any attached files.

**Do you want your identity to be public for this peer review?** For information about this choice, including consent withdrawal, please see our Privacy Policy .

Reviewer #1: No

Reviewer #2: No

Reviewer #3: **Yes: ** Chuan Kok Lim

**Figure resubmission:**

**Reproducibility:**



---

## [Decision Letter · Decision Letter 1]

11 Aug 2025

Response to Reviewers
Revised Manuscript with Track Changes
Manuscript

Shaden Kamhawi

co-Editor-in-Chief

Paul Brindley

co-Editor-in-Chief

**Additional Editor Comments:**
**Reviewers' comments:**

**Key Review Criteria Required for Acceptance?**

**Methods:**

-Are the objectives of the study clearly articulated with a clear testable hypothesis stated?

-Is the study design appropriate to address the stated objectives?

-Is the population clearly described and appropriate for the hypothesis being tested?

-Is the sample size sufficient to ensure adequate power to address the hypothesis being tested?

-Were correct statistical analysis used to support conclusions?

-Are there concerns about ethical or regulatory requirements being met?

Reviewer #2: (No Response)

Reviewer #3: (No Response)

Reviewer #4: Please see my comments

**Results:**

-Does the analysis presented match the analysis plan?

-Are the results clearly and completely presented?

-Are the figures (Tables, Images) of sufficient quality for clarity?

Reviewer #2: (No Response)

Reviewer #3: (No Response)

Reviewer #4: Please see my comments

**Conclusions:**

-Are the conclusions supported by the data presented?

-Are the limitations of analysis clearly described?

-Do the authors discuss how these data can be helpful to advance our understanding of the topic under study?

-Is public health relevance addressed?

Reviewer #2: (No Response)

Reviewer #3: (No Response)

Reviewer #4: Please see my comments

**Editorial and Data Presentation Modifications?**

Reviewer #2: (No Response)

Reviewer #3: (No Response)

Reviewer #4: Please see my comments

**Summary and General Comments:**

Reviewer #2: (No Response)

Reviewer #3: The authors have appropriately addressed the reviewers' comments

Reviewer #4: This is an interesting study to assess the Ab responses to a live JEV vaccine administered to young Philippino children (9-24 months of age) living in an area with endemic circulation of DENVs, JEV, Zika. The investigators conducted the study to support broader use of JEV vaccines in the country and to address concerns about a JEV vaccine enhancing dengue.

The main questions they address consist of

1) Does pre-existing flavivirus immunity suppress or enhance the JEV vaccine response?

2) Does the JEV vaccine stimulate cross reactive binding and functional Abs to dengue type 2?

3) Does the JEV vaccine stimulate antibodies with the potential to enhance DV2 infections

The investigators obtain clear answers to the first 2 questions. They observe that prior flavivirus immunity has no impact on vaccine take or magnitude of the JEV NAb response. They also observe no boosting of DV2 NAbs by the JEV vaccine.

The studies to answer the 3rd question are misguided and poorly interpreted. Cell culture ADE assays have no predictive value for immune enhanced dengue disease. See my comments below.

Finally, while this is a straightforward study, the results are poorly presented, and the reader is left quite confused about the results and their implications. See my comments below.

Specific comments

1) The investigators appear to be unfamiliar with basic aspects of flavivirus serology, and the study is not presented in a proper context. The investigators determine dengue immune status before vaccination by testing for DV2 binding Abs. They observe 8/28 DV2 positive children. For the entire paper they consider these children to be DV2 immune. If they are consistent, they would also call these children JEV immune because 7/8 are also positive for JEV binding Abs. The VLP binding assays do not have the required specificity for determining virus-specific immunity because of extensive cross reactivity. Please use the term flavivirus immune because the binding Abs could be the result of a past exposure to any DV serotype, JEV or Zika. The assay for determining virus specific immunity is the neutralization assay. Only 3/8 children with binding Abs actually neutralized DV2. Please revise the paper to reflect the fact that 8/28 kids were flavi immune at baseline and only 3 of these kids were actually DV2 immune.

2) The finding that nearly all the children respond to the vaccine and develop JEV NAb, with minimal impact on DV2 NAb is important but lost in the way the results are presented. Please consider reorganizing the figures by presenting results relevant to specific questions? In the current paper, many graphs with different comparisons are presented in an indiscriminate manner.

3) The JEV vaccine drives a NAb response in nearly all children, but this is not reflected in the JEV VLP binding assay, where many children remain negative. This could be due to vaccine induced IgM competing with IgG or a technical issue with the assay. Please discuss.

4) As already mentioned, cell culture ADE is not predictive of risk in people. Even the cell culture ADE measurements are confusing. The investigators choose fold enhancement. Why do some sera enhance at the highest concentration tested while others require significant dilution of the serum. If one serum sample enhanced 5 fold at a dilution of 1:20 only and another serum reaches a peak of 5 fold enhancement at a serum dilution of 1:200, as presente din this paper they both have a peak fold enhancement of 5. Are they really the same? If the authors look closely at the ADE curves, they will see that kids with high DV2 neut titers have to be diluted to enhance while those with no NAb sometimes enhance at the highest concertation tested. What does this mean with respect to JEV vaccination enhancing DV2 in vivo? The only message I take from these data are that the cell culture ADE profiles remain the same before and after vaccination suggesting they are being driven by immunity that existed before vaccination and not Abs stimulated by the vaccine.

I hope these comments are useful to reorganize and rewrite this paper to highlight the important results and conclusions of this study.

PLOS authors have the option to publish the peer review history of their article (what does this mean? ). If published, this will include your full peer review and any attached files.

**Do you want your identity to be public for this peer review?** For information about this choice, including consent withdrawal, please see our Privacy Policy .

Reviewer #2: No

Reviewer #3: **Yes: ** Dr Chuan Kok Lim

Reviewer #4: No

**Figure resubmission:****Reproducibility:** To enhance the reproducibility of your results, we recommend that authors of applicable studies deposit laboratory protocols in protocols.io, where a protocol can be assigned its own identifier (DOI) such that it can be cited independently in the future. Additionally, PLOS ONE offers an option to publish peer-reviewed clinical study protocols. Read more information on sharing protocols at https://plos.org/protocols?utm_medium=editorial-email&utm_source=authorletters&utm_campaign=protocols

---

## [Editor Report · Decision Letter 2]

9 Sep 2025

Dear Dr. Vista,

We are pleased to inform you that your manuscript 'Antibody responses to Japanese encephalitis virus and dengue virus serotype 2 in children from an orthoflavivirus endemic region after IMOJEV vaccination' has been provisionally accepted for publication in PLOS Neglected Tropical Diseases.

Best regards,

Joshua Anzinger

Academic Editor

David Safronetz

Section Editor

Shaden Kamhawi

co-Editor-in-Chief

Paul Brindley

co-Editor-in-Chief

---

## [Editor Report · Acceptance letter]

Dear Dr. Vista,

We are delighted to inform you that your manuscript, "Antibody responses to Japanese encephalitis virus and dengue virus serotype 2 in children from an orthoflavivirus endemic region after IMOJEV vaccination," has been formally accepted for publication in PLOS Neglected Tropical Diseases.

Best regards,

Shaden Kamhawi

co-Editor-in-Chief

Paul Brindley

co-Editor-in-Chief
